Relic populations of Fukomys mole-rats in Tanzania: description of two new species F. livingstoni sp. nov. and F. hanangensis sp. nov.

Faulkes Chris G. c.g.faulkes@qmul.ac.uk 1
Mgode Georgies F. 2
Archer Elizabeth K. 1
Bennett Nigel C. 3
1 School of Biological & Chemical Sciences, Queen Mary University of London , London , UK
2 Pest Management Centre, Sokoine University of Agriculture , Morogoro , Tanzania
3 Department of Zoology & Entomology, University of Pretoria , Pretoria , Gauteng , South Africa
Kraatz Brian
Electronic publication date: 2017 Apr 27
Publication date: 2017
Volume: 5
Electronic Location ID: e3214
Received 2016 Oct 6; Accepted 2017 Mar 21
Copyright: ©2017 Faulkes et al.
Copyright year: 2017
Copyright holder: Faulkes et al.
License: This is an open access article distributed under the terms of the Creative Commons Attribution License, which permits unrestricted use, distribution, reproduction and adaptation in any medium and for any purpose provided that it is properly attributed. For attribution, the original author(s), title, publication source (PeerJ) and either DOI or URL of the article must be cited.
License URL: https://creativecommons.org/licenses/by/4.0/

Keywords: Fukomys, Mitochondrial DNA, African mole-rats, Rift Valley, Phylogeography, Bathyergidae, New species

Funding: National Research Foundation University of Pretoria South Africa This work was funded by grants from the National Research Foundation and University of Pretoria South Africa (to NCB). The funders had no role in study design, data collection and analysis, decision to publish, or preparation of the manuscript.

==============================
Previous studies of African mole-rats of the genera Heliophobius and Fukomys (Bathyergidae) in the regions of East and south central Africa have revealed a diversity of species and vicariant populations, with patterns of distribution having been influenced by the geological process of rifting and changing patterns of drainage of major river systems. This has resulted in most of the extant members of the genus Fukomys being distributed west of the main Rift Valley. However, a small number of isolated populations are known to occur east of the African Rift Valley in Tanzania, where Heliophobius is the most common bathyergid rodent. We conducted morphological, craniometric and phylogenetic analysis of mitochondrial cytochrome b (cyt b) sequences of two allopatric populations of Tanzanian mole-rats (genus Fukomys) at Ujiji and around Mount Hanang, in comparison with both geographically adjacent and more distant populations of Fukomys. Our results reveal two distinct evolutionary lineages, forming clades that constitute previously unnamed species. Here, we formally describe and designate these new species F. livingstoni and F. hanangensis respectively. Molecular clock-based estimates of divergence times, together with maximum likelihood inference of biogeographic range evolution, offers strong support for the hypothesis that vicariance in the Western Rift Valley and the drainage patterns of major river systems has subdivided populations of mole-rats. More recent climatic changes and tectonic activity in the “Mbeya triple junction” and Rungwe volcanic province between Lakes Rukwa and Nyasa have played a role in further isolation of these extra-limital populations of Fukomys in Tanzania.

Introduction

African mole-rats of the family Bathyergidae are subterranean rodents that occur throughout sub-Saharan Africa (Faulkes & Bennett, 2013), with much of their range subdivided by the Great Rift Valley. They have been widely studied as a result of the variation in their social and reproductive strategies, and comparative studies have become crucial in this respect (Allard & Honeycutt, 1992; Faulkes et al., 1997; Faulkes & Bennett, 2013). More recently, the naked mole-rat (Heterocephalus glaber) has also emerged as a model species for the study of longevity and cancer resistance (Gorbunova et al., 2014). Hence a clear understanding of their taxonomy, biodiversity and phylogenetic relationships has become increasingly important, not least because they are a speciose group, but also because there are a number of genetically unique, disjunct populations that are limited in their distributional range (Faulkes et al., 2004; Ingram, Burda & Honeycutt, 2004; Van Daele et al., 2007a; Van Daele et al., 2007b). Historically, systematics of the group has been problematic, because cryptic diversity is prevalent in mole-rats due to convergent morphological evolution of a phenotype adapted to the subterranean niche. However, molecular phylogenies based on both nuclear and mitochondrial genes have produced congruent trees (e.g., Allard & Honeycutt, 1992; Faulkes et al., 1997; Walton, Nedbal & Honeycutt, 2000; Huchon & Douzery, 2001; Faulkes et al., 2004; Ingram, Burda & Honeycutt, 2004; Van Daele et al., 2007a; Van Daele et al., 2007b).

Plate tectonics and the formation of the Great Rift Valley have played a central role in the adaptive radiation and distribution of the Bathyergidae, particularly among mole-rats of the genera Heliophobius and Fukomys (Faulkes et al., 2004; Faulkes et al., 2010; Faulkes et al., 2011). These taxa are distributed widely, with virtually all members of the genus Fukomys occurring in locations west of the Western (Albertine) and Southern Rift Valleys from northern South Africa, through south-central Africa to Uganda and Sudan. South-central Zambia in particular is a hot-spot for species/karyotypic diversity in Fukomys, possibly as a result of changing patterns of drainage over geological time (Van Daele et al., 2004; Van Daele et al., 2007a; Van Daele et al., 2007b). Disjunct populations are found in Ghana, Cameroon and Nigeria, and a small number of isolated populations occur east of the Rift Valley in Tanzania, where the silvery mole-rat Heliophobius is widespread and the predominant bathyergid rodent (Faulkes et al., 2011).

Faulkes et al. (2010) investigated a number of populations of Fukomys (or Cryptomys sensu lato) in Tanzania in an attempt to clarify their taxonomic status and to confirm the nomenclature proposed by the earliest reports published by Allen & Loveridge (1933). The latter originally described a new taxon (Cryptomys hottentotus occlusus) from Kigogo in south-western Tanzania, interpreting it as a locally adapted form of Cryptomys hottentotus whytei (Fukomys whytei stricto sensu; Van Daele et al., 2007a), which is geographically the closest in distribution to C. h. occlusus. Allen and Loveridge also report catching F. whytei (stricto sensu) from further north at Ujiji. An additional two, more distant locations (Mount Hanang and Liwale), were later recorded for C. h. occlusus by Swynnerton & Hayman (1951) in their checklist of Tanzanian mammals. The study by Faulkes et al. (2010) concluded that Fukomys whytei constitutes a clear phylogenetic species, supporting the “whytei” clade described by Van Daele et al. (2007a), and that C. hottentotus occlusus (sensu Allen & Loveridge, 1933) should be subsumed into F. whytei or, at most, considered a subspecies. With regard to animals sampled from populations at Liwale and Hanang, the former were found to be Heliophobius rather than Fukomys (Faulkes et al., 2011), while genetic analysis of two mole-rats from Hanang appeared to constitute a previously unrecognised species (Faulkes et al., 2010). At the time it was not possible to obtain samples from the remaining sites at Ujiji described by Allen and Loveridge.

Here, using molecular phylogenetic and morphometric techniques, we characterize and name a new species from the population of mole-rats in the region surrounding Mount Hanang and further north at Mbulu. As Allen & Loveridge (1933) reported mole-rats at Ujiji that have not been studied before, we also collected and analysed samples from this locale, and in doing so describe and name a second new species.

Methods

Sampling, PCR and sequencing

Samples were obtained from three main locations in Tanzania between August 2006 and July 2013 (Ujiji: n = 6, Hanang: n = 9 and Mbulu: n = 31), to compare with other geographically relevant material already collected and sequenced (Faulkes et al., 2004; Van Daele et al., 2007a; Van Daele et al., 2007b; Table 1, Fig. 1). Tissue (muscle or skin biopsies and whole animals) was fixed in 95% ethanol and then stored at −20 °C prior to DNA extraction and/or morphological analysis. Genomic DNA was extracted from the tissue samples and PCR amplification of the entire cytochrome b (cyt b) gene (1,140 bp) carried out using primers and protocols previously described for African mole-rats by Faulkes et al. (1997). Sequencing was carried out in both directions using combinations of primers to obtain complementary partially overlapping strands (20–100% overlap), using the Eurofins Genomics Value Read automated sequencing service (Eurofins Genomics, Ebersberg, Germany).

Table 1 Collection and sample data for F. hanangensis, F. livingstoni and F. whytei, together with Genbank Accession numbers and the respective haplotype (hapl.) for cyt b sequences.

QMUL, Queen Mary University of London, refers to institutionally archived samples.

Species	Sample no	Sample	Location	GPS	Altitude	Colony	Sex	bw	Age	GenBank	cyt b	Specimen	
	QMUL	NHM London	type		Lat	Long	(m)			(g)	class	Accession no	hapl.		
F. hanangensis	3926	–	QMUL	Hanang	S04°24′	E035°27′	na	na	na	na	2	GU197596	A	skull & tissue	
F. hanangensis	3927	–	QMUL	Hanang	S04°24′	E035°27′	na	na	na	na		GU197595	B	tissue only	
F. hanangensis	3928	–	QMUL	Hanang	S04°29′	E035°24′	1964	na	na	na	2	–	–	skull & tissue	
F. hanangensis	4303	NHMUK 2015.14	Paratype	Hanang	S04°25.761′	E035°27.158′	1896	1	M	40		KX905166	B	whole animal	
F. hanangensis	4304	–	QMUL	Hanang	S04°25.761′	E035°27.158′	1896	1	F	50		KX905167	B	whole animal	
F. hanangensis	4305	–	QMUL	Hanang	S04°25.412′	E035°27.453′	1856	2	M	120		KX905168	B	tissue only	
F. hanangensis	4306	–	QMUL	Hanang	S04°25.412′	E035°27.453′	1856	2	na	na		KX905169	B	tissue only	
F. hanangensis	4307	–	QMUL	Hanang	S04°25.412′	E035°27.453′	1856	2	na	50		KX905170	B	tissue only	
F. hanangensis	4308	NHMUK 2015.15	Holotype	Hanang	S04°29.510′	E035°24.519′	1957	1	BrF	62		KX905171	B	whole animal	
F. hanangensis	4309	NHMUK 2015.16	Paratype	Mbulu	S04°3.165′	E035°26.430′	2135	2	F	57	2	KX905172	C	skull & tissue	
F. hanangensis	4310	NHMUK 2015.17	Paratype	Mbulu	S04°2.591′	E035°27.511′	2188	2	F?	80	3	KX905173	C	skull & tissue	
F. hanangensis	4311	NHMUK 2015.18	Paratype	Mbulu	S04°2.591′	E035°27.511′	2188	2	M	68	2	KX905174	C	skull & tissue	
F. hanangensis	4312	NHMUK 2015.19	Paratype	Mbulu	S04°2.591′	E035°27.511′	2188	2	M?	55	2	KX905175	C	skull & tissue	
F. hanangensis	4313	NHMUK 2015.20	Paratype	Mbulu	S04°2.591′	E035°27.511′	2188	2	F	47	1	KX905176	C	skull & tissue	
F. hanangensis	4314	NHMUK 2015.21	Paratype	Mbulu	S04°4.091′	E035°26.668′	2180	3	M	140		–	–	whole animal	
F. hanangensis	4315	NHMUK 2015.22	Paratype	Mbulu	S04°4.091′	E035°26.668′	2180	3	M?	85	2	KX905177	C	skull & tissue	
F. hanangensis	4316	NHMUK 2015.23	Paratype	Mbulu	S04°4.091′	E035°26.668′	2180	3	F	54	1	KX905178	C	skull & tissue	
F. hanangensis	4317	NHMUK 2015.24	Paratype	Mbulu	S04°4.091′	E035°26.668′	2180	3	M?	65	3	KX905179	C	skull & tissue	
F. hanangensis	4318	NHMUK 2015.25	Paratype	Mbulu	S04°3.528′	E035°26.189′	2179	4	F	60		–	–	whole animal	
F. hanangensis	4319	NHMUK 2015.26	Paratype	Mbulu	S04°3.528′	E035°26.189′	2179	4	M	100		–	–	whole animal	
F. hanangensis	4320	NHMUK 2015.27	Paratype	Mbulu	S04°3.528′	E035°26.189′	2179	4	M	103	3	KX905180	C	skull & tissue	
F. hanangensis	4321	NHMUK 2015.28	Paratype	Mbulu	S04°3.528′	E035°26.189′	2179	4	M	85		–	–	whole animal	
F. hanangensis	4322	NHMUK 2015.29	Paratype	Mbulu	S04°3.528′	E035°26.189′	2179	4	F	79	3	KX905181	C	skull & tissue	
F. hanangensis	4323	NHMUK 2015.30	Paratype	Mbulu	S04°3.528′	E035°26.189′	2179	4	F	75	3	KX905182	C	skull & tissue	
F. hanangensis	4324	NHMUK 2015.31	Paratype	Mbulu	S04°3.528′	E035°26.189′	2179	4	M	60	2	–	–	skull & tissue	
F. hanangensis	4325	NHMUK 2015.32	Paratype	Mbulu	S04°3.528′	E035°26.189′	2179	4	M	35	1	–	–	skull & tissue	
F. hanangensis	4326	NHMUK 2015.33	Paratype	Mbulu	S04°2.818′	E035°26.029′	2115	5	M	130	3	KX905183	C	skull & tissue	
F. hanangensis	4327	NHMUK 2015.34	Paratype	Mbulu	S04°2.818′	E035°26.029′	2115	5	M	75	1	KX905184	C	skull & tissue	
F. hanangensis	4328	NHMUK 2015.35	Paratype	Mbulu	S04°2.818′	E035°26.029′	2115	5	M	130		–	–	whole animal	
F. hanangensis	4329	NHMUK 2015.36	Paratype	Mbulu	S04°2.818′	E035°26.029′	2115	5	M?	60	2	KX905185	C	skull & tissue	
F. hanangensis	4330	NHMUK 2015.37	Paratype	Mbulu	S04°2.818′	E035°26.029′	2115	5	na	53	1	KX905186	C	skull & tissue	
F. hanangensis	4331	NHMUK 2015.38	Paratype	Mbulu	S04°2.818′	E035°26.029′	2115	5	M	65	3	KX905187	C	skull & tissue	
F. hanangensis	4332	NHMUK 2015.39	Paratype	Mbulu	S04°2.818′	E035°26.029′	2115	5	M	35	2	–	–	skull & tissue	
F. hanangensis	4333	NHMUK 2015.40	Paratype	Mbulu	S04°3.505′	E035°25.853′	2203	6	F	100	2	KX905188	C	skull & tissue	
F. hanangensis	4334	NHMUK 2015.41	Paratype	Mbulu	S04°3.505′	E035°25.853′	2203	6	M	140	4	–	–	skull & tissue	
F. hanangensis	4335	–	QMUL	Mbulu	S04°3.505′	E035°25.853′	2203	6	F	75		–	–	whole animal	
F. hanangensis	4336	–	QMUL	Mbulu	S04°3.793′	E035°26.294′	2163	7	M	55	1	KX905189	C	skull & tissue	
F. hanangensis	4337	–	QMUL	Mbulu	S04°3.793′	E035°26.294′	2163	7	F	87		–	–	whole animal	
F. hanangensis	4338	–	QMUL	Mbulu	S04°3.793′	E035°26.294′	2163	7	M	130	3	KX905190	C	skull & tissue	
F. hanangensis	4339	–	QMUL	Mbulu	S04°3.793′	E035°26.294′	2163	7	M	115	3	KX905191	C	skull & tissue	
F. livinstoni	5208	NHMUK 2015.42	Holotype	Ujiji	S04°51.760′	E028°42.326′	2601	1	M	50	2	KX905192	D	whole animal	
F. livinstoni	5209	NHMUK 2015.43	Paratype	Ujiji	S04°51.760′	E028°42.326′	2601	1	F	35	1	KX905193	D	skull & tissue	
F. livinstoni	5210	NHMUK 2015.44	Paratype	Ujiji	S04°51.693′	E029°42.335′	2624	2?	F	80	4	KX905194	D	skull & tissue	
F. livinstoni	5211	NHMUK 2015.45	Paratype	Ujiji	S04°51.701′	E029°42.340′	2620	3?	M	42	1	KX905195	D	skull & tissue	
F. livinstoni	5212	NHMUK 2015.46	Paratype	Ujiji	S04°51.760′	E028°42.326′	2601	1	M	38	2	KX905196	D	whole animal	
F. livinstoni	5213	NHMUK 2015.47	Paratype	Ujiji	S04°51.620′	E029°41.542′	2601	4	M	52	2	KX905197	E	whole animal	
F. whytei	3913	–	QMUL	Kigogo	S08°37.905′	E035°12.2754′	1970	1	F	101	1	GU197597	W-A	skull & tissue	
F. whytei	3915	–	QMUL	Kigogo	S08°38.3442′	E035°11.7912′	1946	2	M	134	3	GU197599	W-B	skull & tissue	
F. whytei	3916	–	QMUL	Kigogo	S08°38.1516′	E035°12.5676′	na	3	F	124	3	GU197600	W-A	skull & tissue	

Figure 1 Map showing the relative locations of the main Fukomys clades considered in this study as defined by Van Daele et al. (2007a), Faulkes et al. (2004), Faulkes et al. (2010) and Faulkes et al. (2011), with letters in circles corresponding as follows (see Fig. 2): u, Ujiji; h, Hanang; L, Lufubu clade; W, West Bangweulu clade; E, East Bangweulu clade; v, F. vandewoestijineae; m, F. mechowii; a, F. amatus; mi, F. micklemi; an, F. anselli; d, F. damarensis; k, F. kafuensis; dr, F. darlingi. F. bocagei populations are located further west beyond the coverage of this map as indicated.

Red circles correspond to the F. whytei clade (F. whytei and F. whytei occlusus), the geographically closest known populations of Fukomys to Hanang and Ujiji (Faulkes et al., 2010). Red circles with a white dot denote the origin of F. whytei samples included in the skull shape analysis. The area encircled by the broken line corresponds to the Mbeya triple junction fault and the Rungwe volcanic province, linking the Rukwa and Malawi rifts. Lower panels show detailed sampling maps for Ujiji (left) and Hanang (right), indicating the individual catching sites and relative distribution of cyt b haplotypes. Map data: (A) Google, DigitalGlobe, (B) Landsat.

Ethical note

Animals were euthanised with an overdose of chloroform on the day of capture. Sexing, weighing and tissue collection were carried out post mortem. Fieldwork was funded and approved by the University of Pretoria, Animal Use & Care Committee Approval EC053-09. Sampling focused around agricultural areas where mole-rats are considered pests. Collection permits were issued by the Sokoine University of Agriculture and respective District Authorities (Hanang, Mbulu and Ujiji). The Tanzania Commission for Science and Technology (COSTECH) granted a research permit for collection of rodents (permit no. 2013-260-NA-2014-110) to Dr. Georgies Mgode. Export permits were obtained from the Wildlife Department (Ministry of Natural Resources and Tourism Tanzania), and Zoosanitary/Veterinary permit from the Ministry of Livestock Development and Fisheries.

Analysis of mitochondrial DNA sequences

Sequences were aligned manually for analysis using Mesquite version 3.03 (Maddison & Maddison, 2014) and phylogenetic relationships investigated using parsimony, maximum likelihood and Bayesian approaches. These methodologies build evolutionary trees using different underlying assumptions and algorithms, and it is usual to investigate the data with all three approaches. Parsimony aims to produce a tree that reflects the minimum number of evolutionary changes needed to explain the data, while likelihood and Bayesian methods take a probabilistic approach based on models of molecular evolution. Maximum Likelihood fits of 24 different nucleotide substitution models were used to establish the evolutionary model most appropriate for the data (from Hierarchical Likelihood Ratio tests), and these parameters were then used in subsequent analyses, where appropriate. Maximum likelihood and parsimony analyses were undertaken and phylogenetic trees and genetic distances among haplotypes based on nucleotide sequences constructed using MEGA 6 (Tamura et al., 2013). Maximum likelihood was conducted using the heuristic search option, with initial tree(s) for the search obtained automatically by applying the Maximum Parsimony method. For maximum parsimony we used the min-mini heuristic algorithm with a search factor of 1 with gaps treated as missing data and eliminated from the analysis. Bootstrap analysis was conducted with 100 replicates of the dataset.

Bayesian phylogenetic analysis was undertaken using BEAST v1.8.2 (Drummond & Rambaut, 2007; Drummond et al., 2012). Following the molecular clock likelihood ratio test performed using MEGA 6 (Tamura et al., 2013) to establish the correct molecular clock model, the null hypothesis of equal evolutionary rate throughout the tree was rejected (likelihood ratio = 9.92; P > 0.001). Thus an uncorrelated relaxed molecular clock model (Drummond et al., 2006) and a Yule tree prior (the most suitable for interspecies comparisons) were selected in BEAST, and an HKY model of molecular evolution. The molecular clock rate was calibrated by assuming a divergence time of 10–11 Mya for the common ancestor of Cryptomys/Fukomys (the ingroup in this study), and these divergence times for the ingroup were input as a prior with upper (11) and lower (10) limits. This calibration has been previously used by Ingram, Burda & Honeycutt (2004), Van Daele et al. (2007a) and Faulkes et al. (2010), and was based on a timing of 19 Mya for the divergence of the Heliophobius lineage within the bathyergid family tree, and the occurrence of the fossil Proheliophobius (Lavocat, 1973). After initial data exploration with independent chains we implemented a final run having a chain length of 30,000,000, sampling output every 30,000 iterations. The first 300,000 trees (10%) were discarded during burn-in. Mixing and convergence of MCMC chains generated by BEAST were investigated and checked using Tracer v1.6.0 to ensure sufficient iterations and sampling were performed before samples from the posterior distribution of trees were summarized using Treeannotator v1.8.2, and trees drawn using FigTree v.1.4.2 (Drummond et al., 2012).

Each distinct haplotype obtained from the two geographical locations (Ujiji and Hanang) were included in all phylogenetic analyses, together with the published sequences representative of the main clades of Fukomys (Van Daele et al., 2007a; Faulkes et al., 2010), and two other bathyergid mole-rats as outgroups: Cryptomys hottentotus hottentotus and Heliophobius emini (Faulkes et al., 2011). In addition, a previously unpublished Fukomys sequence from Ghana was included as another example of an extralimital, but geographically distant population.

Phylogeographic analysis

Phylogeographic analysis and hypothesis testing of the ingroup clade Cryptomys and Fukomys was performed using likelihood analysis of geographic range evolution, utilizing the dispersal-extinction-cladogenesis (DEC) model implemented in the program Lagrange v.2.0.1 (Ree & Smith, 2008). Within the Lagrange DEC model, the cladogenesis parameter remains fixed, while the other two parameters (dispersal/range expansion and extinction/range contraction) are estimated. Seven geographical areas were defined, based on the distribution of extant clades: south of the Zambezi River (SZ); south of the Limpopo River (SL); west of the Kafue River (WK); east of the Kafue River (EK); Mbeya Triple Junction (MTJ); east of the Rift Valley (ER) and West Africa (WA). Three models were then implemented as follows: The unconstrained null model (M0) allows geographic ranges to include any combination of areas (with input range and dispersal matrices in Lagrange left as the default values of 1.0). M1 restricts ranges to include a maximum of four adjacent areas, as follows: WA/EK/WK, ER/MTJ/EK, MTJ/EK/WK/SZ, EK/WK/SZ/SL, WK/SZ/SL, and SZ/SL, but does not restrict dispersal. The third more complex (stratified) model restricts ranges as in M1, and also constrains dispersal differentially across four time periods, 0–2, 2–5, 5–8 and 8–12 Mya, according to proposed changing patterns of drainage and physical barriers resulting from rifting. Thus, for the period 12–8 Mya, we constrain dispersal to occur only to and from regions SZ and SL, corresponding to the emergence of the common ancestor of Cryptomys/Fukomys in southern/South Africa (Faulkes et al., 2004; Ingram, Burda & Honeycutt, 2004). For the period 8–5 Mya, in our model dispersal is more relaxed as Fukomys spreads north (from SL and SK) through south central and east (WK, EK, MTJ and ER) to West Africa. Movement back again from WA to all regions, and from ER and MTJ to WA is not permitted during this phase (the latter reflecting proposed barriers formed by rifting); movement south from all regions to SL is prohibited except from SZ. As rifting and drainage evolution of major river systems (particularly the Zambezi and Kafue) is hypothesized to become more significant between 5–2 Mya (Van Daele et al., 2007a; Van Daele et al., 2007b; Moore, Cotterill & Eckardt, 2013), we further increase constraints on dispersal to also prohibit movement from SL and SZ north to WA, WK, EK, MTJ and ER. For the period 2–0 Mya and the final phases of rifting and drainage evolution, dispersal is only permitted from ER to MTJ, between MTJ and EK, from WK to SZ and SL, and between SZ and SL.

Morphology, craniometrics and analysis of skull shape

Pelage colour was recorded under natural daylight by consensus of three observers, with reference to Munsell Soil Color Charts (1954 Edition; Munsell Color Co., Inc. Baltimore, USA). Subsequent descriptions of colour all refer to this scale. Morphometric measurements were taken from a total of 34 skulls (Hanang/Mbulu region, n = 26; Ujiji, n = 5; Fukomys whytei, n = 3) using digital callipers (to the nearest 0.1 mm), as described by Verheyen, Colyn & Hulselmans (1996) (Fig. S1), together with standard head, body, tail and hind foot length measurements, in specimens where the entire body was available. Age classes were estimated from tooth eruption and wear characteristics as previously described for Fukomys damarensis (Bennett, Jarvis & Wallace, 1990).

Differences in craniometric measurements were investigated with MANOVA. Any measurements that were shown to have significant differences were then tested using a separate univariate ANOVA with a Tukey HSD to ascertain which species were statistically different. Bonferroni corrections were applied to these results to account for multiple testing. Because there were only three skulls available from F. whytei for craniometric measurements, this small sample size could potentially bias the statistical analysis. Therefore, a second MANOVA was carried out with just F. hanangensis and F. livingstoni measurements. A third MANOVA on just F. hanangensis craniometric measurements (the largest sample size) enabled us to look for sex and age class related differences in skull morphology. All ANOVA-based analyses were carried out using the ‘stats’ package in R 3.2.3 (R Core Team, 2015).

In order to investigate and quantify any differences in skull morphometrics between Fukomys livingstoni and Fukomys hanangensis, shape variation was investigated using landmark analysis, as previously described in Faulkes et al. (2010). The dorsal and ventral surfaces of skulls from specimens collected at Hanang (n = 2), Mbulu (n = 24), Ujiji (n = 5), and Kigogo (F. whytei, the geographically closest species; n = 3), were photographed three times to minimize the effects of misalignment. For further comparison, material (n = 20) from F. whytei was obtained in the form of photographs of the dorsal and ventral surfaces of the skulls collected by Allen & Loveridge (1933), and were provided by the Harvard Museum of Comparative Zoology (see Faulkes et al., 2010 for further information). In addition, photographs of the dorsal and ventral surfaces of the skulls from the more geographically remote F. anselli (n = 20) from Lusaka, Zambia were also included (this species is part of the micklemi clade in the genetic analysis). The relative locations of these samples are displayed in Fig. 1. To capture the shape, the 2-D coordinates of a total of 15 dorsal and 17 ventral landmarks as previously described in Faulkes et al. (2010; Fig. S2) were placed on each photograph and digitized using the TpsDIG2 software (Version 1.4; Rohlf, 2004), and mean relative warp scores for each specimen produced by tpsRelW (Version 1.36; Rohlf, 2003) were plotted.

New zoological taxonomic names

The electronic version of this article in Portable Document Format (PDF) will represent a published work according to the International Commission on Zoological Nomenclature (ICZN), and hence the new names contained in the electronic version are effectively published under that Code from the electronic edition alone. This published work and the nomenclatural acts it contains have been registered in ZooBank, the online registration system for the ICZN. The ZooBank LSIDs (Life Science Identifiers) can be resolved and the associated information viewed through any standard web browser by appending the LSID to the prefix http://zoobank.org/. The LSID for this publication is: [urn:lsid:zoobank.org:pub:DC6D5104-CB60-48A1-9A06-B16A25DC6573]. The online version of this work is archived and available from the following digital repositories: PeerJ, PubMed Central and CLOCKSS.

Results

General capture information

Ujiji

A total of six animals were sampled from two sites 1.5 km apart and at an altitude of 2,601 to 2,624 m above sea level, at Msimba Village on the outskirts of Ujiji (Table 1; Fig. 1). The capture sites were either in or directly adjacent to fields containing maize, sweet potato, cassava, palms and bananas. At one location at Site 1 (Msimba village, Kasaka hamlet), an adult male and young female were captured from the same trap (NHMUK 2015.42 and NHMUK 2015.43), with an adult male (NHMUK 2015.46) trapped a few metres away and likely from the same burrow. An adult female and a young male (NHMUK 2015.44 and NHMUK 2015.45) were caught at further locations nearby (115 m distant) at Site 1 and may represent different colonies. All individuals had the same cyt b Haplotype “D”. At Site 2 (Msimba village, Kabemba site), a single adult male (NHMUK 2015.47) was caught at 2,601 m above sea level in a valley with sweet potato fields and uncultivated grassland. This individual was assigned to a different cyt b Haplotype “E”.

Hanang/Mbulu

At Hanang, 640 km east and slightly north of Ujiji, nine animals were collected from two sites around Hanang Mountain at altitudes of 1,856 to 1,957 m above sea level: at Gitting Village and at Jorodom Village, 8 km SW from Gitting (Table 1; Fig. 1). The capture sites were either in uncultivated grassland adjacent to or on the edge of fields next to cultivated mixed crops, bananas and planted trees. A single individual each had Haplotype “A” or Haplotype “B” at Gitting, while the remaining six at Gitting and two at Jorodom Village had Haplotype “B” (Table 1; Fig. 1).

Further north thirty-one animals were collected at altitudes of 2,115 to 2,188 m above sea level from seven distinct catching sites/colonies from another population at Mbulu, 42 km north of Gitting Village (Hanang). Habitat at Mbulu was similar to that around Hanang, either in uncultivated grassland adjacent to or on the edge of fields next to cultivated mixed crops (e.g., potatoes, maize, sugarcane, and bananas). From one to eight individuals were caught at the seven colony sites (Table 1). All twenty of the thirty-one animals sequenced from the Mbulu population had the same Haplotype “C”.

Phylogenetic relationships

The maximum likelihood tree with the highest log likelihood (−5146.172) is shown in Fig. 2A. Initial trees for the heuristic search were obtained automatically by applying the maximum parsimony method. A discrete Gamma distribution was used to model evolutionary rate differences among sites (5 categories (+G, parameter = 0.453)). The rate variation model allowed for some sites to be evolutionarily invariable ([+I], 52.275% sites; Tamura & Nei, 1993; Tamura et al., 2013). Bootstrap support was high for the main taxonomic groupings (80–100%), with the exception of Lufubu (69%), although the latter was consistently placed as the sister group to the East Bangweulu clade in all trees.

Figure 2 (A) Phylogenetic relationships based on maximum likelihood analysis of 25 cytochrome b (cyt b) mitochondrial DNA ingroup haplotypes and two outgroups: Heliophobius and Cryptomys hottentotus hottentotus. Clade descriptors and circular symbols correlate with maps in Figs. 1 and 2, while the numbers at each node on the branch refer to the percentage bootstrap values following 100 replications; (B) differences in topology (indicated by red lines) of the four equally parsimonious trees produced from maximum parsimony analysis, for the clade indicated by the symbol • in (A). Haplotypes labeled for F. livingstoni and F. hanangensis sp. nov. correspond to those cited in the text, Fig. 2 and Table 1, other species are designated according to current taxonomic understanding and GenBank Accession Numbers (Van Daele et al., 2007a; Van Daele et al., 2013; Faulkes et al., 2010).

Figure 3 Maximum clade credibility tree inferred using BEAST with an uncorrelated relaxed molecular clock model (allowing a variable rate of sequence evolution across the tree).

The clock calibration to convert genetic distance to time is based on calibration 1 (Ingram, Burda & Honeycutt, 2004). The tree is also annotated with maximum likelihood reconstruction of geographic range evolution under the constrained (stratified) DEC model implemented in Lagrange (see text). Additional geological annotations below the tree are based on Voje et al. (2009), Trauth et al. (2005) and Ebinger et al. (1993), respectively. MTJ = corresponds to the period of increased tectonic activity at the Mbeya Triple Junction. Circled letters and the filled red circle on clade labels correspond to locations on Fig. 1, with GenBank accession numbers in parentheses after species names and locations. Geographic ranges of these extant taxa are denoted with abbreviations and colour coded boxes, and are defined below. Blue bars spanning nodes correspond to ages for the lower and upper bound of the 95% highest posterior density (HPD) intervals. Numbers at nodes refer to the posterior probabilities in support of that node (1 = 100%). Node A corresponds to the divergence of F. livingstoni, and Node B the divergence of F. hanangensis. The temporal position of these nodes, together with their HPD intervals, for clarity are further represented by the vertical dotted lines and the blue bars below the tree and above the geological annotation bar respectively The colour filled squares at nodes represent ancestral ranges for diverging lineages reconstructed by Lagrange. The matrices above the tree associated with four time ranges depict possible area-to-area dispersals, with rows corresponding to “from” and columns “to”, green fill “yes” and no fill “no”. Areas as follows: SZ, south of the Zambezi River; SL, south of the Limpopo River; WK, west of the Kafue River; EK, east of the Kafue River; MTJ, Mbeya Triple Junction; ER, east of the Rift Valley; WA, West Africa. Relative positions of these geographic areas are depicted on the map inset, and follow the same abbreviations and colour codes as the filled squares on the tree nodes and taxon labels. Blue lines depict the Zambezi (Z), Kafue (K) and Limpopo Rivers (L), blue patches lakes, and bold black lines the main arms of the Rift Valley. Note that for clarity, other rivers are omitted.

Maximum parsimony analysis produced four trees of length 883 (Fig. 2B; consistency index = 0.466; retention index = 0.612). From the total of 1,140 characters, 674 were constant, 325 variable characters were parsimony informative and 141 uninformative. One of the trees (Fig. 2B (i)) was identical to the maximum likelihood tree. The other three most parsimonious trees differed in the relative placement of the whytei clade with respect to Lufubu, East Bangweulu, West Bangweulu and amatus clades, and the swapping of lineages of F. whytei from Mbala and Kigogo within the whytei clade (Fig. 2B (ii), (iii) and (iv)).

Bayesian phylogenetic analysis performed with BEAST produced a tree identical to maximum likelihood tree 2a and maximum parsimony tree 2b (i), with all nodes having high posterior support (0.86 to 1.00/86–100%; Fig. 3).

Maximum likelihood, maximum parsimony and Bayesian trees were all congruent in supporting previously accepted taxonomic groupings (whytei, Lufubu, East Bangweulu, West Bangweulu, F. amatus, F. darlingi, F. damarensis, F. micklemi, F. bocagei, F. mechowii, and F. vandewoestijneae; Van Daele et al., 2007a; Van Daele et al., 2007b; Faulkes et al., 2010). The sequence of F. zechi from Ghana constitutes an extant representative of an early diverging lineage in the Fukomys clade, supporting previous studies that extra-limital populations of Fukomys in West and central Africa represent relic populations from an initial radiation of ancestral Fukomys (Ingram, Burda & Honeycutt, 2004). Next a clade containing F. mechowii, F. bocagei and the recently described F. vandewoestijineae (Van Daele et al., 2013) constitute a group of species distributed through central and west central Africa (Zambia and Angola), with an earlier common ancestor to the populations found at the geographically distant Ujiji. All trees consistently placed the animals collected from Ujiji and Hanang/Mbulu in reciprocally monophyletic groups, separated by the divergence of a major clade containing F. darlingi, F. damarensis, and F. micklemi/F. kafuensis. Finally, a monophyletic group containing five distinct clades was consistently recovered (West Bangweulu, F. amatus, Lufubu, East Bangweulu and F. whytei; Fig. 2).

The phylogenetic analysis provides evidence for two hitherto unrecognised phylogenetic species in Tanzania, one from Ujiji (Fukomys livingstoni sp. nov.) and a second from the Hanang/Mbulu region (Fukomys hanangensis sp. nov.). Both were genetically divergent from one another within the molecular phylogeny for the Fukomys genus, and also from the geographically closest clade containing F. whytei. Each of the Fukomys sp. nov. comprise clades strongly supported by bootstrap values of 100% (maximum likelihood) and posterior probabilities of 1.00 (Bayesian trees).

Table 2 Mean cyt b genetic distances between sequences for each haplotype (%).

Below diagonal are uncorrected p distances, above diagonal Tamura-Nei + Gamma (1.4964) corrected rates of substitutions.

	1	2	3	4	5	6	7	8	9	10	11	12	13	14	15	16	
1. whytei clade	–	4.3	4.8	6.3	5.9	7.3	9.5	9.8	10.3	11.6	12.5	12.3	11.8	18.9	25.9	30.4	
2. Lufubu clade	4.0	–	3.4	5.0	5.1	7.0	8.2	9.2	10.2	10.0	12.7	10.9	11.0	19.6	23.8	29.4	
3. East Bangweulu clade	4.5	3.3	–	5.7	5.4	7.2	9.0	9.8	9.9	10.7	12.8	11.2	11.0	20.3	23.8	30.5	
4. West Bangweulu clade	5.7	4.6	5.2	–	5.7	8.4	9.3	11.3	10.7	11.1	13.7	11.3	11.4	19.7	25.7	31.0	
5. F. amatus clade	5.4	4.7	4.9	5.2	–	7.9	9.7	10.2	11.2	9.8	12.6	11.2	11.7	20.0	24.9	29.9	
6. F. hanangensis	6.5	6.3	6.4	7.4	7.0	–	8.2	8.5	9.1	9.9	13.7	12.1	11.9	18.5	24.3	31.6	
7. F. darlingi	8.2	7.3	7.9	8.0	8.3	7.3	–	7.6	8.0	10.3	14.2	11.7	10.7	18.5	27.4	31.3	
8. F. damarensis	8.5	8.0	8.5	9.6	8.8	7.5	6.8	–	7.1	11.1	12.9	12.8	13.2	19.5	25.1	28.6	
9. F. micklemi clade	8.8	8.8	8.5	9.1	9.5	8.0	7.1	6.4	–	10.5	14.1	13.4	12.9	19.7	27.1	29.7	
10. F. livingstoni	9.7	8.6	9.1	9.4	8.5	8.5	8.8	9.5	8.9	–	13.3	11.4	11.3	18.8	24.8	28.5	
11. F. bocagei	10.5	10.7	10.7	11.3	10.5	11.3	11.6	10.7	11.5	11.0	–	9.3	10.2	22.5	22.4	30.1	
12. F. mechowii	10.3	9.3	9.6	9.6	9.6	10.1	9.9	10.7	11.0	9.7	8.1	–	4.8	21.2	29.5	32.2	
13. F. vandewoestijneae	9.9	9.3	9.3	9.6	9.8	9.9	9.1	10.9	10.5	9.5	8.8	4.5	–	21.1	26.9	31.7	
14. F. zechi	14.6	15.1	15.5	15.1	15.3	14.5	14.5	15.1	15.1	14.5	16.6	16.0	15.8	–	28.3	31.8	
15. Outgroup 1	18.2	17.3	17.3	18.1	17.8	17.7	19.0	18.0	18.8	17.8	16.8	20.0	18.7	19.7	–	30.7	
16. Outgroup 2	21.5	21.1	21.6	21.8	21.4	22.2	22.0	20.8	21.3	20.5	21.5	22.4	22.1	22.3	21.6	–	

Inter-clade sequence divergence

Maximum Likelihood fits of 24 different nucleotide substitution models indicated that the Tamura-Nei + G + I (TN93+G+I) model of sequence evolution was the most appropriate. Both mean uncorrected-p and TN93+G+I corrected genetic distances between lineages/clades represented in Figs. 2 and 3 are displayed in Table 2. Uncorrected p (and (TN93+G+I) distances between F. hanangensis and ingroup lineages from different locations ranged from a minimum of 6.3% (7.0%) versus the Lufubu clade to 14.5% (18.5) versus F. zechi. Uncorrected p distances between F. livingstoni and ingroup lineages ranged from a minimum of 8.5% (9.8) versus F. amatus to 14.5% (18.8) versus F. zechi. Mean p distance between F. livingstoni and F. hanangensis was also 8.5% (9.9), while p distances between these and the geographically closest F. whytei clade were 9.7% (11.6) and 6.5% (7.3) respectively, exceeding the distances among some currently recognised species, e.g., F. amatus and F. whytei 4.7% (5.9%). Genetic distances between haplotypes within F. hanangensis and F. livingstoni clades were very low. For the three F. hanangensis haplotypes (A, B and C) the total number of substitutions over the 1,140 bp (and uncorrected p distances) were A versus B: 1 (0.001%), A versus C: 4 (0.004%), and B versus C: 3 (0.003%). For the two F. livingstoni haplotypes D and E the total number of substitutions over the 1,140 bp (and uncorrected p distances) were: 6 (0.007%).

Molecular clock estimates of divergence times and phylogeographic analysis

Figure 3 summarises the molecular clock-based divergence times, together with 95% highest posterior density (HPD) intervals (equivalent to 95% confidence intervals), for the main nodes within the phylogeny generated using BEAST. According to the maximum clade credibility tree produced by BEAST (Fig. 3) the divergence of the Ujiji lineage and F. livingstoni (Node A in Fig. 3) occurred in the Pliocene at 3.55 Mya, with a 95% HPD extending into the Early Pleistocene (representing 2.63 to 4.89 Mya). Following the common ancestor of the F. darlingi, F. damarensis and F. micklemi clades with species now extant in south central Africa, the divergence of the Hanang/Mbulu lineage and F. hanangensis (Node B in Fig. 3) is estimated at 2.36 Mya in the Pleistocene (lower and upper bound of the 95% HPD = 1.68–3.25 Mya). Thus the timings of divergence of both the Ujiji and Hanang/Mbulu lineages may precede the commencement of increased tectonic activity at the Mbeya Triple Junction (MTJ; Fig. 3), which forms the conduit between south central Africa and Tanzania. A sister group to F. hanangensis contains five clades with taxa restricted to Zambia, with the exception of the F. whytei clade (red circles in Fig. 1), which includes lineages that diverged much more recently in the Pleistocene. Geographically these populations of F. whytei are concentrated around the MTJ region, with only Kigogo population significantly within Tanzania.

Assuming a confidence window of two log-likelihood units (Ree & Smith, 2008), Lagrange analysis of historical biogeography showed successively more likely scenarios when transitioning from the null (M0) to the more complex M1 and stratified models. This strongly supported our phylogeographic hypothesis, with the latter having a greater likelihood score (less negative) than the alternative M0 and M1 models (by 6.91 and 3.94 log-likelihood units respectively; Table 3). From the null model, estimates of the rate of dispersal successively increase while estimates of the rate of extinction decline. Thus, in the stratified model, the range data are better explained by a model of dispersal than a model of extinction.

Table 3 Summary of inferences from Lagrange analysis of ancestral area and range evolution, under DEC models of historic biogeography.

M0 represents the unconstrained (null) model that allows geographic ranges to include any combination of areas. M1 restricts ranges to include a maximum of three adjacent areas. The stratified model constrains dispersal differentially across four time periods, according to changing patterns of drainage and physical barriers resulting from rifting (Fig. 3).

Model	Log likelihood score (−lnL)	Rate of dispersal	Rate of extinction	
M0	−44.83	0.02183	0.00798	
M1	−41.86	0.03791	4.285e−09	
Stratified	−37.92	0.0878	8.287e−09	

Figure 3 incorporates the results from Lagrange for the stratified model, and shows the optimal maximum likelihood reconstruction of geographic range evolution. This supports the hypothesis that from the common ancestor of Cryptomys and Fukomys, there was a dispersal north and range expansion into south-central and West Africa in Fukomys, with Cryptomys restricted to areas south of the Zambezi River (SZ). The lineages that follow the divergence of F. zechi into West Africa (WA) then show a pattern of cladogenesis with ancestral ranges for the descendant lineages encompassing Tanzania, east of the Rift Valley (ER) and across Zambia (EK and WK). Later (more recent) dispersal south has given rise to populations of F. darlingi and F. damarensis south of the Zambezi, and also south of the Limpopo (SL) in the latter. Recent dispersal into Tanzania into the ER region appears to be blocked, resulting in populations in the Mbeya Triple Junction area (MTJ).

Morphometric analysis of skulls

Morphometric analysis of F. livingstoni and F. hanangensis, together with F. whytei from south western Tanzania and F. anselli from Lusaka, Zambia, differentiated skulls in line with the proposed taxonomy, when relative warps 1 and 2 from either or both the dorsal and ventral surfaces were plotted (Figs. 4A and 4B). For example, the dorsal surface analysis clearly separates F. livingstoni and F. anselli from each other, and from an unresolved cluster of points from F. hanangensis and F. whytei skulls. The ventral surface differentiates all four taxa, although there is a small overlap of points between F. livingstoni and F. hanangensis. The overall skull shape changes occurring along the relative warp axes are captured in thin plate spline plots in Fig. 4A (dorsal surface) and Fig. 4B (ventral surface).

Figure 4 Scatter plot of sample means of relative warp 1 (x-axis) against relative warp 2 (y-axis) from the shape analysis, together with thin plate splines showing landmark displacements from the consensus dorsal skull surface (the origin) to the extent of the diagonal plotted in each quadrant (e.g., upper left −0.14, 0.08 in (A)) for (A) dorsal, and (B) ventral skull surfaces.

Plot symbols squares = males, circles = females, diamonds = sex unknown, F. hanangensis animals from Hanang. Colours: orange, F. livingstoni; black, F. hanangensis; grey, F. whytei; dark purple, F. anselli. Circled F. hanangensis outlier is animal 4332, a very small 35 g male (Table S1).

The clear separation of the F. livingstoni samples from the consensus shape (corresponding to the origin in the plot in Fig. 4A) and the F. hanangensis/F. whytei morphospace, along relative warps 1 and 2 of the dorsal surface is due to three main effects: (i) anterior-medial shifting of the jugal within the zygomatic arch (landmarks 4 and 6), (ii) shortening of the nasal bones, particularly at their posterior extent (landmarks 1, 2, 14 and 15), and (iii) anterior/anterior-medial shifts in the parietal (landmarks 8, 9 and 12; Fig. S2 and Fig. 4A). To some extent these same changes occur when moving from the F. hanangensis/F. whytei morphospace (i.e., upper right quadrant of Fig. 4A) to F. anselli (upper left quadrant of Fig. 4A), as this is also a shift along the x-axis. The additional separation of F. livingstoni from F. anselli points along the y-axis result from a posterior-lateral shifts in the narrowest inflection of squamosal (landmark 10) and the right anteriolateral tip of the parietal bone (landmark 9), and a posterior-medial shift in the anterior tip of the interparietal bone (landmark 11). A single small (35 g) male F. hanangensis is an outlier to the main cluster of points for this species, perhaps not being fully developed cranially.

On the ventral skull surface, changes in F. livingstoni from the consensus shape (and F. hanangensis) are principally due to a small lateral shift in the posterior tip of auditory bulla at the junction with the occipital (landmark 9) and a small anterior-medial shift in the junction of the jugal and zygomatic process (landmark 4). These changes are present, but less exaggerated, in F. whytei, which also occupies the morphospace in the upper right quadrant of the relative warp plot (Fig. 4B). Changes from the consensus shape in the main group of F. hanangensis points were small as they cluster quite close to the origin on the plot. The separation of F. anselli along the y-axis results from anterior shifts in both the junction of squamosal and auditory bulla, and the lateral tip of auditory bulla (landmarks 7 and 8), and a posterior shift in the junction of jugal and zygomatic process (landmark 4; Fig. 4B).

Intraspecific differences in shape between the sexes was not apparent on either the dorsal or ventral plots for F. whytei, F. hanangensis or F. anselli, with points for males and females intermingled. With only a single female skull for F. livingstoni it is not possible to draw conclusions, although on both dorsal and ventral plots the female separates from the main group of male samples.

The full dataset of standard craniometric measurements of skulls are displayed in Table S1. Despite the small sample sizes for F. whytei and F. livingstoni, MANOVA analysis together with F. hanangensis revealed an overall global significant difference among the three species (Pillai’s Trace = 1.84, F2,28 = 3.59, P = 0.01), and significant differences in thirteen of the twenty-three measurements (summarized in Table S2). However, individual ANOVA on each of these significant measurements, followed by post-hoc Tukey tests with Bonferroni correction to investigate among species differences, left only one measurement that was significantly different (P < 0.05), due to the effects of the correction (for multiple testing) and small sample size for F. whytei. This metric was the greatest breadth of first upper molar (M13), with F. whytei larger than both F. hanangensis and F. livingstoni (2.30 ± 0.07 vs 1.90 ± 0.02 and 1.87 ± 0.04 mm respectively).

MANOVA analysis of just F. hanangensis and F. livingstoni, revealed a global significant difference among the two species (Pillai’s Trace = 0.99, F1,27 = 15.04, P < 0.01), with the following seven measurements that were individually identified as significantly different (mean ± SEM values for F. hanangensis and F. livingstoni respectively): (i) greatest length of skull (M1), F1,27 = 4.30, P = 0.04, 35.19 ± 0.51 vs 32.27 ± 0.84 mm; (ii) length of diastema (M6), F1,27 = 6.89, P = 0.01, 10.70 ± 0.26 vs 8.70 ± 0.34 mm; (iii) smallest interorbital breadth (M8), F1,27 = 4.36, P = 0.04, 8.87 ± 0.23 vs 7.78 ± 0.03 mm; (iv) zygomatic breadth on the zygomatic process of the squamosal (M9), F1,27 = 5.59, P = 0.03, 24.01 ± 0.47 vs 21.08 ± 0.72 mm; (v) breadth of upper dental arch (M12), F1,27 = 8.31, P = 0.01, 5.43 ± 0.04 vs 5.18 ± 0.11 mm; (vi) greatest length of nasals (M16), F1,27 = 9.41, P < 0.01, 12.22 ± 0.25 vs 10.15 ± 0.49 mm, and (vii) mediosagittal projection of rostrum height at anterior border of first upper molars (M22) F1,27 = 5.18, P = 0.03, 10.05 ± 0.22 vs 8.57  ± 0.31 mm. These differences indicate that, compared with F. livingstoni, the skull of F. hanangensis is longer and wider at the zygomatic arch, with a rostrum that is longer and higher, but not broader. These differences support the geometric morphometric shape analysis in areas where the landmarks also capture the craniometric measurements. For example, the anterior-medial shifting of the jugal within the zygomatic arch (dorsal landmarks 4 and 6, ventral landmark 4) reflects the change in zygomatic breadth, while the shortening of the nasal bones, at their posterior extent (landmarks 1, 2, 14 and 15) reflects the difference in rostrum length.

Because of the small sample size (n = 5) of F. livingstoni a comprehensive statistical analysis of craniometrics among sexes and age classes was not possible for this species. A MANOVA analysis of F. hanangensis including sex and age class as predictors revealed no significant differences in skull craniometrics between the sexes (Pillai’s Trace = 0.80, F1,19 = 15.04, P = 0.87) and among age classes (Pillai’s Trace = 0.93, F1,19 = 1.43, P = 0.54). The lack of sex differences in craniometric measurements is also evident in the shape analysis plots in Fig. 4, where the male and female points for F. hanangensis are intermingled.

Description of species

Family Bathyergidae Waterhouse, 1841	
Genus FukomysKock et al., 2006	
Fukomys livingstoni sp. nov.	
Livingstone’s mole-rat (common name)	
LSID urn:lsid:zoobank.org:act:67DEACE5-3163-4FAE-885B-8EC04F072EEC	

Holotype

NHMUK 2015.42 is an adult male collected in July 2013 at the Kasaka hamlet within the village of Msimba, near Ujiji. The specimen is composed of a skin and skull in very good condition (Figs. 5A and 5B; Fig. 6A). The external measurements (mm) are: head and body length 115.4, tail 8.9 and hind foot 22.0 (Table 1). The body weight was 50 g. The pelage is darkish grey brown overall with a shorter very dark grey under-fur and a small irregularly shaped light grey head patch.

Paratypes

A further five specimens (NHMUK 2015.43 – NHMUK 2015.46) were collected from around the type locality (Table 1; Fig. 1).

Etymology

This species is named after Dr. David Livingstone, as Ujiji (the type locality) is the site of the famous meeting on 10 November 1871 when Henry Morton Stanley found the explorer David Livingstone, who many thought to be dead, and uttered the famous words “Dr. Livingstone, I presume?” (Stanley, 1872).

Type locality

Msimba village, 6.4 km northeast from the city centre of Ujiji (S04°51.760′; E029°42.326′). The specimen was trapped in a valley at an altitude of 793 m (2,601 ft) above sea level, in an area with moist sandy soil, where cassava, sweet potato, maize, palms and bananas were being cultivated.

Distribution and biology

The full range of this species remains to be determined with collection of the series described here restricted to around the village of Msimba on the outskirts of Ujiji. The holotype was captured from the same hole in the burrow as a young adult female (NHMUK 2015.43), with an adult male (NHMUK 2015.46) trapped a just few metres away and probably from the same burrow. The presence of adults and young adults together in a burrow suggests natal philopatry and cooperative breeding that is a characteristic of species within the genus Fukomys.

Figure 5 Dorsal (A) and (C) and ventral (B) and (D) views of holotypes: Fukomys livingstoni (5208/NHMUK 2015.42) (A) and (B) and Fukomys hanangensis (4308/NHMUK 2015.15), (C) and (D).

Scale bar = 1 cm.

Figure 6 Skulls of (A) Fukomys livingstoni sp. nov. (5208/NHMUK 2015.42; holotype), and (B) Fukomys hanangensis sp. nov. (4334/NHMUK 2015.41; paratype) in dorsal, ventral and lateral view, and mandible in lateral and dorsal view.

Diagnosis

Individuals of this species may be clearly distinguished from adjacent populations of F. hanangensis and F. whytei on the basis of morphology and molecular (DNA sequence) data. Morphologically, F. livingstoni is smaller with a mean adult body size (age class 2 and above) of 55 ± 8.9 g (n = 4) compared with F. hanangensis (mean adult body weight = 83.4 ± 5.6 g; n = 30; Table 1). Compared with F. hanangensis, the skull of F. livingstoni is shorter and narrower at the zygomatic arch, with a rostrum that is shorter and with less height at the mediosagittal projection at the anterior border of the first upper molars, but not different in breadth (Table S1 ; Fig. 6). A head spot (bles) is present in the specimens obtained for this study, although the presence/absence of the bles may not always be a reliable diagnostic feature as there is often specific variation in other bathyergids.

Description (and comparison with other species)

This is a small species of Fukomys: the four adults (age classes 2 to 4) ranged from 38–80 g in body weight of (mean = 55.0 ± 8.9 g; Table 1), similar in proportion to F. darlingi found in Zimbabwe, where mean body weights are 65.3 ± 14.1 g (males) and 62.9 ± 14.9 (females; Bennett, Jarvis & Cotterill, 1994). Sexual dimorphism in F. livingstoni remains to be investigated fully as there was only a single young female skull in our sample (age class 1; Table S1). When compared with other species of Fukomys from south-central Africa, the ratio of body length to body weight and the size of the skull (expressed as greatest width/greatest length) is at the lowest end of the distribution, and much smaller than F. damarensis and F. mechowii (Fig. S4; Kingdon et al., 2013). Geometric morphometric analysis of the dorsal skull surface revealed differences between F. livingstoni and F. anselli, F. hanangensis and F. whytei. This is manifest as an anterior-medial shifting of the jugal within the zygomatic arch, shortening of the nasal bones, and anterior/anterior-medial shifts in the parietal. This results in the shorter narrower skull described in the diagnosis above, although not at the expense of rostrum width. On the ventral skull surface, changes observed in F. livingstoni were less pronounced, and principally due to a small lateral shift in the posterior tip of auditory bulla at the junction with the occipital and a small anterior-medial shift in the junction of the jugal and zygomatic process. Overall pelage colouration varied from darkish grey-brown to brown and dark brown, with shorter under-fur of very dark grey or black. A small irregularly-shaped head spot was present, varying from light grey to white in colour (Table S2; Fig. 5A). In this respect F. livingstoni is similar to neighbouring F. whytei, whose range extends into south-western Tanzania, where a small head spot is reported to be present in some populations (Allen & Loveridge, 1933; Burda et al., 2005). Otherwise F. livingstoni is much smaller than F. whytei, where body weight means (g) for animals from the type locality of Karonga, Malawi were 132.7 ± 22.3 (males, n = 4) and 121.5 ± 10.7 (females, n = 4; Burda et al., 2005). Specimens of F. whytei geographically closer to Ujiji (from Kigogo, Tanzania) were also within this size range, with a young animal of age class 1 (and 3/4 cheek teeth erupted) recorded at 101 g, larger than adult F. livingstoni (Table 1). To the trained eye, pelage colour may also distinguish the more grey-brown/brown F. livingstoni from F. whytei, although the latter is reportedly variable among populations from cinnamon-grey to dark slatey (Allen & Loveridge, 1933) and grey-buff (Burda et al., 2005). Body size also clearly distinguishes F. livingstoni from the larger F. hanangensis (see below) found further north in Tanzania, and while the latter lacks a paler coloured head spot (at least in the sample set reported here) and tends to be more yellowish brown, there is some overlap in pelage colouration.

Family Bathyergidae Waterhouse, 1841	
Genus FukomysKock et al., 2006	
Fukomys hanangensis sp. nov.	
The Hanang mole-rat (common name)	
LSID urn:lsid:zoobank.org:act:59C00958-9628-461F-987D-AB897F52598F	

Holotype

NHMUK 2015.15 is an adult breeding female, collected in September 2009 from Jorodom village on the slopes of Mount Hanang. The specimen is composed of an entire body preserved in ethanol in very good condition (Figs. 5C and 5D). The external measurements (mm) are: head and body length 111.0, tail 12.2 and hind foot 23.1 (Table 1). The body weight was 62 g. The pelage is brown overall with a shorter black under-fur. No head spot is present.

Paratypes

A further 39 specimens including 27 paratypes (NHMUK 2015-14 and NHMUK 2015.16 – NHMUK 2015.41) and 12 samples retained at Queen Mary, University of London for further analysis. Eight of these were collected from around the type locality at Hanang, while the remaining 31 were from locations around Mbulu (Table 1; Fig. 1).

Etymology

Named after the location where the specimens were first collected around Mount Hanang, Tanzania.

Type locality

Jorodom village, (S04°29.510′; E035°24.519′). The specimen was trapped in a valley at an altitude of 1,957 m (6,422 ft) above sea level, in an uncultivated grass field surrounded by crop fields.

Distribution and biology

Currently the range of this species is known to be around Mount Hanang including the villages of Gitting and Jorodom, and extending to at least 40 km further north to Mbulu. The full range of this species remains to be determined. Aside from the first three animal captures in 2006, the remaining 37 specimens collected in 2009 from Gitting, Jorodom and Mbulu were gathered from 10 colonies with up to eight being caught from one burrow (Colony 4 at Tumati-Eyasirong, Mbulu; Table 1). These are not maximum colony sizes as burrows were not completely trapped out, and no breeding females were captured at Mbulu. Specimens from Colony 4 consisted of five males and three females, including a young 35 g male of age class 1, and mature adults of age classes 2 and 3. A similar spread of age classes was also seen among the animals collected from Mbulu Colony 5. These observations suggest natal philopatry and cooperative breeding for this species.

Diagnosis

Individuals of this species may be clearly distinguished from adjacent populations of F. livingstoni and the more geographically distant F. whytei on the basis of morphology and molecular (DNA sequence) data. Morphologically, F. hanangensis is larger than neighbouring F. livingstoni, with a mean adult body size (i.e., excluding animals of known age class 1) of 83.4 ± 5.6 g (range: 35–140; n = 30; Table 1). Compared with F. livingstoni, the skull of F. hanangensis is longer and wider at the zygomatic arch, with a rostrum that is longer and higher, but not broader (Table S1; Fig. 6).

Description (and comparison with other species)

F. hanangensis is a small to medium sized example of the genus Fukomys, while at an average adult size of 83 g it is larger than F. livingstoni (mean adult body weight = 55.0 ± 8.9 g, range: 38–80 g; n = 4; Table 1), it is slightly smaller in proportions to F. whytei, where body weight means (g) for animals from the type locality of Karonga, Malawi were 132.7 ± 22.3 (males, n = 4) and 121.5 ± 10.7 (females, n = 4; Burda et al., 2005). In comparison with other species of Fukomys from south-central Africa (Fig. S4), the ratio of body length to body weight and the size of the skull are smaller than F. damarensis and F. mechowii. However, it would be hard to distinguish F. hanangensis from species such as F. darlingi, F. anselli, F. bocagei, F. kafuensis and F. vandewoestijneae on the basis of body size alone. There was a trend towards male F. hanangensis being larger than females: 90.8 ± 8.0 g (n = 19) versus 72.5 ± 4.8 g (n = 10) respectively, although this was not significant (P = 0.127, t = 1.575, df = 27). However, there is no sexual dimorphism evident in either skull shape or craniometrics. Geometric morphometric analysis of the dorsal skull surface revealed no shape differences between F. hanangensis and F. whytei. However, when compared with F. anselli and F. livingstoni, there is a posterior-lateral shifting of the jugal within the zygomatic arch, lengthening of the nasal bones, and posterior/posterior-lateral shifts in the parietal. On the ventral skull surface, differences between F. hanangensis and F. livingstoni, F. whytei and (to a lesser extent) F. anselli, are more pronounced. This is manifest as a posterior shift in the posterior border of the molars, a small posterior-lateral shift in the junction of the jugal and zygomatic process, and a change in shape of the auditory bulla. The latter arises from posterior lateral shifts in both the junction of squamosal and auditory bulla, and the lateral tip of auditory bulla, and a small medial shift in the posterior tip of auditory bulla at the junction with the occipital. Overall pelage colouration varied from yellowish brown through dark yellowish brown to brown/dark brown. Under-fur was normally black, with two specimens very dark grey. None of the series described here had a lighter-coloured head spot present. Thus F. hanangensis can be morphologically distinguished from neighbouring F. livingstoni by its larger size and a more yellowy brown coat than F. livingstoni, and a lack of lighter head spot.

Discussion

This study builds on preliminary sequence data from two mole-rats collected at Hanang in Tanzania, originally reported by Faulkes et al. (2010). Not only do we confirm the presence of a previously unrecognised species of African mole-rat in this region (Fukomys hanangensis), but also provide robust evidence for a second new species from specimens collected from a new locale at Ujiji (Fukomys livingstoni). We base our descriptions of the new species primarily on genetic data, although clear morphological differences are also evident in geometric morphometric analysis of skull shape, craniometrics, pelage and body size. Within the genus Fukomys, both F. hanangensis and F. livingstoni are distinct evolutionary lineages, as defined by the Phylogenetic Species Concept (Cracraft, 1989), and form separate clades nested among other clearly defined species in the cyt b molecular phylogeny. An increasing number of single locus mtDNA (in particular cyt b, but also 12S rRNA) phylogenies have produced robust and consistent phylogenies for the Bathyergidae and clarified the taxonomy of cryptic species (Allard & Honeycutt, 1992; Faulkes et al., 1997; Faulkes et al., 2004; Faulkes et al., 2010; Faulkes et al., 2011; Walton, Nedbal & Honeycutt, 2000; Huchon & Douzery, 2001; Ingram, Burda & Honeycutt, 2004). Recent phylogenomic analysis of the main lineages (genera) within the family using data from 3,999 concatenated genes agree fully with single gene studies, producing a tree with a congruent topology (Davies et al., 2015). Single nuclear gene-based studies (e.g., intron 1 of the nuclear transthyretin (TTR) gene) have also been useful, but such loci have been shown to be far less variable that mitochondrial cyt b, and thus lack phylogenetic signal and resolving power (see Ingram, Burda & Honeycutt, 2004). While Ingram et al. found that their TTR tree was not significantly different to their mitochondrial-based tree, the latter had more phylogenetic structure and the TTR data was unable to resolve most of the sub-clades within Fukomys. Importantly, it could not resolve such clearly defined species as F. damarensis, F. darlingi and F. kafuensis/micklemi, and therefore we did not sequence nuclear genes for this study.

The previously unpublished sequence for F. zechi (the Togo mole-rat) from Ghana, which we also include in our phylogeny here, formed a distinct and highly divergent lineage within the Fukomys clade as a whole. This would place F. zechi with the other poorly known extra-limital West and Central African Fukomys as relic populations of the initial radiation of the genus, including F. foxi from Cameroon and F. ochraceocinereus from South Sudan (Ingram, Burda & Honeycutt, 2004).

The divergent nature of F. hanangensis and F. livingstoni within the topology of the cyt b gene tree is also evident in the magnitude of the genetic distances between the clades, adding further support for their evolutionary distinctiveness (and within clade differences between haplotypes were very low). For example corrected p distances between geographically adjacent F. whytei versus F. hanangensis and F. livingstoni are 7.3 and 9.7% respectively, while F. hanangensis and F. livingstoni differ by 8.5%. These values exceed those of recognised species within the Fukomys clade, for example F. amatus versus F. whytei (5.9%), and F. mechowii versus F. vandewoestijneae (4.8%; Table 2).

We have previously drawn attention to the potential role of rifting and volcanic activity in cladogenesis within Fukomys (Faulkes et al., 2004; Faulkes et al., 2010), while Van Daele et al. (2004), Van Daele et al. (2007a) and Van Daele et al. (2007b) hypothesise that consequent shifts in the patterns of drainage of major river systems in south-central Africa occurring in the Pliocene/Pleistocene have further subdivided populations of mole-rats. Our phylogeographic hypothesis testing using likelihood analysis of geographic range evolution firmly supported this scenario. Formation of the African Great Rift Valley began about 50 million years ago (Mya), pre-dating the hypothesised origin of the family Bathyergidae. Later, major rifting occurring in the Miocene, which continued through the Pliocene and Pleistocene and produced the great African lakes, mountains and volcanoes that characterize East Africa (for recent reviews see Chorowicz, 2005 and Macgregor, 2015). An area of particular importance to the radiation of Fukomys and the new species described in this study are the Western and Southern Rifts. This includes Lakes Tanganyika, Rukwa and Nyasa (Malawi), and the corridor of land between them connecting Zambia, Malawi and Tanzania, geologically known as the Mbeya Triple Junction (MTJ; Fig. 1). From at least the early-Pliocene onwards, this area may have constituted the only route for dispersal of terrestrial and subterranean animals, as the Lake Tanganyika basin is thought to have filled to produce a deep lake 6–12 Mya (Cohen, Soreghan & Scholz, 1993), while initial rifting in northern lake Nyasa commencing in the Messinian (upper-most Miocene, 7.2–5.3 Mya; Macgregor, 2015) and attained deep-water conditions by 4.5 Mya (Delvaux, 1995; Delvaux et al., 1998). However, the extent and timing of lake formation in these major rift basins is still controversial, with more recent dates being suggested (see Weiss, Cotterill & Schliewen, 2015 for recent discussion in the context of Lake Tanganyika cichlids).

In our molecular phylogenetic analysis, there was no consistent geographical structuring of the main clades, with geographically adjacent clades divergent in the gene trees. For example, F. hanangensis and F. livingstoni while geographically relatively close are separated from one another and nearby F. whytei by clades endemic to Zimbabwe (e.g., F. darlingi), Zambia, Botswana, Namibia and South Africa (e.g., F. damarensis), and Zambia (taxa within the East and West Bangweulu clades; Figs. 2 and 3). This suggests a series of temporally distinct radiations, perhaps also involving some local extinctions and replacements. In particular, F. livingstoni appears to be a lineage that dispersed into East Africa in the Pliocene (5.33–2.63 Mya), within a large clade with a common ancestor at Node A in Fig. 3 (the 95% HPD values for Node A are 4.89–2.58 Mya, according to the maximum clade credibility tree produced by BEAST). F. mechowi, F. bocagei and F. vandewoestijneae form an immediate outgroup to F. livingstoni, all of which are now distributed further south and west in Zambia, and into Angola in the case of F. bocagei (Fig. 1). F. hanangensis represents either a later, second incursion into Tanzania from a common ancestor occurring 3.25–1.68 Mya (Node B in Fig. 3), or the descendant from that common ancestor which also dispersed into Zambia. While Lakes Tanganyika and Nyasa may have formed a deep water barrier before the divergence of the F. hanangensis lineage and around the time of the earliest estimate for F. livingstoni, the dispersal route from south central Africa to East Africa was possible in the terrestrial corridor between Lakes Rukwa and Nyasa, that now forms the MTJ and Rungwe volcanic province (Branchu et al., 2005; Macgregor, 2015; see Fig. 1). The MTJ/Rungwe volcanic region is at the intersection of the Livingstone basin that forms the north east extremity of Lake Nyasa, the Rukwa–Songwe basin at the south east extremity of the Rukwa rift and the Usangu rift basin. While rifting and volcanism started in this area about 8.6 Mya and intra-basinal faulting, uplift and volcanism were particularly important in shaping the geology of the region at approximately 2.5 Mya, much of the modern topography was generated from 2 Mya and still continues to the present (Ebinger et al., 1993; Delvaux et al., 1998; Mortimer et al., 2007; Macgregor, 2015). Importantly, the range of timings for divergence of the common ancestors of both F. livingstoni and F. hanangensis (4.89–2.63 and 3.25–1.68 Mya respectively; Fig. 3) thus may precede the commencement of increased tectonic activity from 2.5 Mya to the present at the MTJ. It therefore seems highly likely that this increased faulting and uplift contributed to the separation of the south central populations of Fukomys and East African F. hanangensis and F. livingstoni, and today this mountainous habitat represents a significant physical barrier to dispersal for a subterranean rodent, with several points (e.g., Rungwe mountain) exceeding 2,900 m above sea level. Interestingly, F. whytei populations that diverged a little later are focussed mainly around the MTJ, with the exception of the Kigogo population having an earlier common ancestor that is slightly further east in Tanzania, perhaps arriving when dispersal was easier.

The phylogeographic analysis of F. hanangensis and newly acquired F. livingstoni samples add further support for our earlier assertions (Faulkes et al., 2010) that tectonic activity, climatic fluctuations and subsequent expansion and contraction of forest during the Pliocene-Pleistocene may have also played a role in the sub-structuring of populations and cladogenesis in Fukomys. The accompanying Pliocene expansion of C4 grasslands and the savannah habitat in this part of Africa, favoured by mole-rats, would likely have further facilitated range expansion of ancestral populations. The apparently localised and limited distribution of F. hanangensis and F. livingstoni in Tanzania makes assessment of their conservation status and other aspects of their biology a priority.

Supplemental Information

Figure S1 Craniometric points used to take linear measurements of the skull

Craniometric points used to take linear measurements of the skull, with numbers referring to the position of the caliper jaws when taking the measurement as follows: M1 greatest length of skull (between anteriormost point of the incisor to the posteriormost point of the supraoccipital processes); M2 condylobasal length (least distance from the posteriormost projections of the exoccipital condyles to a line connecting the anteriormost projections of the premaxillary bones); M3 henselion-basion length (distance between the posterior margin of the palate and the posteriormost margin of the alveolus of the incisors); M4 henselion-palation length (distance between the anteriormost part of the foramen magnum and the posteriormost margin of the alveolus of the incisors); M5 length of palatal incisive foramen ( = palatal foramen) not recorded; M6 length of diastema (distance between the anterior border of the alveolus of M1 and the posterior border of the alveolus of the upper incisor); M7 distance between the anterior border of the alveolus of M1 and the foremost edge of the upper incisor; M8 smallest interorbital breadth; M9 zygomatic breadth on the zygomatic process of the squamosal; M10 smallest palatal breadth between first upper molars; M11 length of upper cheekteeth (distance between the anterior border of the alveolus of M1 and the posterior border of the alveolus of M4); M12 breadth of upper dental arch: greatest breadth across first upper molars; M13 greatest breadth of first upper molar; M14 smallest breadth of zygomatic plate: distance taken in a plane parallel to the occlusal surface of the upper molar-row; M15 greatest breadth of nasals; M16 greatest length of nasals; M17 length of lower cheekteeth: distance between the anterior border of the alveolus of M1 and the posterior border of the alveolus of M4; M18 greatest breadth of the choanae; M19 length of auditory bulla (the protruding part of the bony Eustachian tube); M20 greatest breadth of braincase; M21 depth of upper incisors (perpendicular on length axis of tooth); M22 mediosagittal projection of rostrum height at anterior border of first upper molars; M23 greatest rostrum breadth (in front of zygomatic plates); M24 distance between the extreme points of the coronoid and the angular processes of the mandibular. Based on Verheyen, Colyn & Hulselmans (1996).

Click here for additional data file.

Figure S2 Positions of landmarks used in the shape analysis

Positions of landmarks used in the shape analysis of the dorsal (a) and ventral (b) skulls. Dorsal landmarks: 1, distal tip of the median line; 2, anterior junction of nasal and premaxilla; 3, junction of rostrum and zygomatic process; 4, junction of jugal and zygomatic process; 5, orbital junction of zygomatic process and frontal; 6, posterior junction of jugal with squamosal; 7, orbital junction of frontal and squamosal; 8, intersection of parietal, squamosal and frontal; 9, right anterolateral tip of the parietal bone; 10, point of narrowest inflection of squamosal viewed from above; 11, anterior tip of the interparietal bone; 12, cross point between the median line and the line which connects left and right anterolateral tip of the interparietal bone; 13, posterior tip of premaxilla; 14, posterior junction of nasal and premaxilla; 15, posterior junction of nasals. Ventral landmarks: 1, distal tip of the premaxilla at the midline; 2, ; lateral extent of the premaxilla at the incisor. 3, anteriormost section of the zygomatic process at the junction of the premaxilla; 4, junction of jugal and zygomatic process; 5, posterior junction of jugal with squamosal; 6, posterolateral edge of the squamosal; 7, junction of squamosal and auditory bulla; 8, lateral tip of auditory bulla; 9, posterior tip of auditory bulla at the junction with the occipital; 10, lateral extent of the foramen magnum; 11, anterior mid-point of the foramen magnum; 12, anterior edge of the occipital at the midline; 13, junction of occipital, basioccipital and auditory bulla; 14, anterior mid-point of the choanae ; 15, posterior b

Click here for additional data file.

Figure S3 Artist’s impressions (drawn from specimens)

(a) Fukomys livingstoni and (b) Fukomys hanangensis. Artwork by Rebecca Gelernter (www.nearbirdstudios.com), who retains the copyright on this image (used with permission). Not to scale.

Click here for additional data file.

Figure S4 Scatterplots showing comparative measurements

(a) Craniometric (greatest skull width at zygomatic arch against greatest skull length) and (b) morphometric data (body length against body weight) for a range of south-central African Fukomys species. Individual points are means with horizontal and vertical error bars indicating sample ranges. Species and sample sizes as indicated (squares denote males, circles females, diamond symbol, sexes unknown). All data from Kingdom et al. (2013) except F. vandewoestinjneae (Van Daele et al., 2013).

Click here for additional data file.

Table S1 Morphometric and craniometric data

Individual and mean ± SEM (mm) head and body length (H&B), tail, and hind-foot lengths, (Hft) together with craniometric measurements M1 to M24, as in Figure S1.

Click here for additional data file.

Table S2 Summary statistics for craniometrics

Results are displayed for both three species (F. livingstoni, F. hanangensis and F. whytei) and two species (F. livingstoni vs F. hanangensis) MANOVA. Significant differences are indicated in bold, and values remining significant after the post-hoc Tukey test and Bonferroni correction indicated as “yes” in the column “Signif.”. Diff refers to the difference in absolute values of the respective measurement (mm) between the first and second species listed (see Table S1).

Click here for additional data file.

We are most grateful to Louise Tomsett at the Natural History Museum, London, for help with museum curation and sample preparation, to Kwaku Dakwa for providing the sample from Ghana, to Dr. Dave Hone for comments on drafts of the manuscript, and to Dr. Steve Le Comber for many helpful discussions and advice on statistical analysis. Thanks also to Judith Chupasko at the Harvard Museum of Comparative Zoology (Department of Mammalogy) for providing photographs of the F. whytei skulls collected by Allen & Loveridge (1933), and Dr. Marietje Oosthuizen for photographing F. anselli skulls collected by Alfred Sichilima. Many thanks also to Lorna Faulkes and Ray Crundwell for other photography. Sequence data is deposited in GenBank with Accession numbers as listed in Table 1.

Additional Information and Declarations

Competing Interests

Author Contributions

Animal Ethics

Field Study Permissions

DNA Deposition

Data Availability

New Species Registration

The authors declare there are no competing interests.

Chris G. Faulkes conceived and designed the experiments, performed the experiments, analyzed the data, contributed reagents/materials/analysis tools, wrote the paper, prepared figures and/or tables, reviewed drafts of the paper.

Georgies F. Mgode and Elizabeth K. Archer performed the experiments.

Nigel C. Bennett conceived and designed the experiments, contributed reagents/materials/analysis tools, reviewed drafts of the paper.

The following information was supplied relating to ethical approvals (i.e., approving body and any reference numbers):

Fieldwork was funded and approved by the University of Pretoria, Animal Use & Care Committee Approval EC053-09.

The following information was supplied relating to field study approvals (i.e., approving body and any reference numbers):

The Tanzania Commission for Science and Technology (COSTECH) granted a research permit for collection of rodents (permit no. 2013-260-NA-2014-110) to Dr. Georgies Mgode.

The following information was supplied regarding the deposition of DNA sequences:

The cytochrome b sequences are accessible via GenBank accession numbers KX905166 to KX905198.

The following information was supplied regarding data availability:

The raw data has been supplied as Table S1.

The following information was supplied regarding the registration of a newly described species:

Species names: urn:lsid:zoobank.org:act:59C00958-9628-461F-987D-AB897F52598F, urn:lsid:zoobank.org:act:67DEACE5-3163-4FAE-885B-8EC04F072EEC, Publication LSID: urn:lsid:zoobank.org:pub:DC6D5104-CB60-48A1-9A06-B16A25DC6573.

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
