# Peer review of "Relic populations of Fukomys mole-rats in Tanzania: description of two new species F. livingstoni sp. nov. and F. hanangensis sp. nov"

_PeerJ, doi:10.7717/peerj.3214_

## Round 0.1 · original submission · Major Revisions

I believe the scientific merit of your study is clear, and the additional sampling and and analyses helps to clarify the systematics and the African genus Fukomys. However, I share several concerns with each of the reviewers.

The overlying issue pointed out from both reviewers is the lack of morphometric sampling for other Fukomys species. I appreciate that the genetic aspects of this study strongly help to distinguish your two novel species (although, please note comments on this by reviewer #2), but your morphometric portion is less clear largely based on low sampling sizes. It seems critical to increase the morphometric sample size of F. whytei, but also to include several more closely related species. Two this point, it is unclear at times what samples were used (or not used) for your analyses, and some clarification is need in the text in terms of previous studies and in the supplemental data table (which has no descriptive title, either). As reviewer 1 points out, the morphometric description should also be more broadly related to other species in your descriptive systematics sections. I know it's a challenge to collect more data such as this, so I do not make this recommendation flippantly.

As pointed out by reviewer #1, there is some overlap in language from the earlier Faulks et al. (2010) that should be corrected.

Here are several other, more minor comments:

1. The callout satellite imagery in figure 2 are not informative.
2. There is a lot of negative space in figure 7; consider scaling the two species to different sizes to their respective morphology can be better assessed.
3. You should include morphologic descriptions of the land marks used in Figure S2.

I very much look forward to seeing a revised version of this work.

Reviewer 1 ·

Basic reporting

This article adheres to most of the PeerJ policies. The writing is clear and provided sufficient background on the history of research in this area with appropriate references. Though not necessary, Figure 1 could use more color coding on map to make it easier to distinguish between different species. Taken on its own, I would say the article is publishable but my major concern with the acceptance of this paper in its current state is the similarity to their previous publication, Faulkes et al 2010 Cladogenesis and endemism in Tanzanian mole-rats, genus Fukomys: (Rodentia: Bathyergidae): a role for tectonics?
Some of the text appears to be taken directly from their previous article (quick comparisons of abstract reveal major similarities), updating the text and analyses with the additional samples. I caution the authors on this pattern and recommend that authors revise the article to focus on what new information adds/changes/supports, removing some of the redundancy with their previous publication.

Experimental design

The research question of this submission is relevant and meaningful, characterizing and naming novel species and understanding the phylogeographic history of this cryptic group of rodent. Using the phylogenetic species concept, they support the recognition of two new species. There are areas for improvement in the writing and analyses.
1) Sampling: While it is difficult and costly to get these animals, the numbers are biased between the groupings and limited on how representative they are of a species. The analyses of this small of a data set provide little power, potentially introducing bias.
2) Morphometrics: The analyses seems appropriate, but what is striking to me is how limited the results are by the sample size. Similar publication that have used morphometrics as a means to differentiate species of mammals have at least 3X the number included for F. hanangensis and over 10X what is presented for F. livingstoni. The choice of additional representatives is questionable. It would be useful to have the colonies/localities identified for F. hanangenesis samples to reveal how points are distributed. Why did they only include the 3 individuals from Kingogo and not the other whytei samples included previously? While they present that these are from the nearest population, the sample size is not appropriate and I would argue they should use all available data. This lack of inclusion that had previously revealed no difference is suspect. Why not include additional species of Fukomys that are closely related to explore the mopho space more thoroughly?
2) Molecular data: this is based on a single locus of mtDNA. While cytb has been used in the past, that was of a time when sequencing was difficult this is an area of debate on whether of not this should be the only line of evidence for recognizing new species. It is a new age of molecular techniques and there are many more markers available, including the 3,999 concatenated genes that they point out have been used for this group. While it may not change the results, the addition of some nuclear loci would provide additional evidence for their hypothesis. It is unclear in the text if they were able get the entire cytb fragment for all individuals sequenced and this should be clarified. If not, missing data should be addressed as this could potentially bias branch lengths and divergence time estimates.
3) Phylogenetic analyses: Since all methods used here are heuristic searches, it is suggested that multiple runs within analysis type (MP, ML, BA) use replicates of each using different starting points, in the form of random seeds, multiple chains of heated vs cold, to determine that all searches are converging on the same topology/tree space. While this may have been done, it isn't clear in the current draft. Authors should clarify or expand on. It is also recommended that >100 bootstraps be performed (1000, 10000).

Validity of the findings

This study provides confirmation of the placement of F. zechi which had previously lacked any molecular data and the phylogenetic placement of two disjunct populations of Fukomys in Tanzania. Phylogenetic relationship within this region (extending to Zambia) have been problematic. While they are able to distinguish these two regions as being distinct lineage with high bootstrap support and posterior probabilities, the relationships to other species of Fukomys and their phylogeographic history is where they lack support (ML BS = 67, 75), not being resolved by this single locus. While this does provide a working hypothesis, it is limited in how much it add to our understanding of evolution within this group. Additional lines of evidence should be included (eg. chromosomal, nuclear).
As stated in the previous section, I find the validity of their findings are limited by sampling. As they state themselves, bathyergid rodents are know for having a great deal of intraspecific variation so a much larger sample size should be included to eliminate the bias a small sample can introduce. Because of this variation, I don't think it is appropriate to only compare skulls of a few individuals.
As they state themselves, they have already provided a detailed interpretation of the

Additional comments

262-263 tone down, evidence isn't overwhelming and robust. Based on a single locus.

301- 303 More evidence, additional loci are needed to document adaptive radiations.

451 - 452 limited dataset, change confirm to support or suggest.

480 remove unequivocal

Reviewer 2 ·

Basic reporting

This article is well structured, but figures and supplementary data should be reorganised. Diagnoses and descriptions should be slightly improved and expanded considering all the Fukomys species. Some suggestions appear in the pdf file.

Experimental design

The scientific approach is rigorous and accurate especially regarding phylogenetic and morphometric methods. However, the authors did not explain clearly the interest of using these different methods, which finally lead to the same results (e.g. parsimony vs maximum likelihood for phylogeny, morphometrics vs basic measurements).

Validity of the findings

The data are discussed according to the geographical proximity of the species (i.e. in Tanzania), but the phylogenetic proximity is not importantly consider especially regarding morphological and morphometric analyses. It is however necessary to integrate this aspect in the diagnoses and the descriptions, to more accurately consider morphological differences between species, and to define the new species on a fair basis (in addition to molecular evidence). As a result, more comparisons with other Fukomys species are needed in the part dealing with systematics. There are also much more to say about the morphological differences between the new species. Morphometric results also need to be confirmed with statistical tests. Genetic distances between the different haplotypes of the new species should also be shown.

Additional comments

If these new data on Tanzanian mole-rats deserve to be published, the author should improve some methodological aspects, as well as diagnoses and a few minor points.
Additional comments and suggestions are detailed in the pdf file.

Annotated reviews are not available for download in order to protect the identity of reviewers who chose to remain anonymous.

---

## Round 0.2 · Minor Revisions

The exchange between you and the reviewers has been useful, I believe, but there are still several lingering issues that need to be addressed.

Please respond to all of the reviewers comments – there were a few inadvertently missed in this last round. I offer up several items here that are largely echoing the reviewers.

(1) The description of shape via your thin plate spines should be incorporated into your species description – there’s useful information there.

(2) I would consider putting either the ventral or dorsal thin plate spines into your figure showing relative warp score plots in the primary text (like you did in the supplemental figures).

(3) You display sex data in figure 4, please comment on that in text.

(4) I realize that the transformations used to get PWS do not allow for significance tests of shape space, but you can do multivariate ANOVAs on your linear data to test for significant differences among species variations.

(5) I am not insisting you include non-cyt-B data, but reviewer 1 makes good points with regard to different trees derived from different genes, and especially in terms of previous studies foci on genera or species, and the implications with regard to the genetics. Please discuss this more clearly in the text, and specifically address differences among taxonomic resolutions.

(6) I also agree that you can better test phylogeographic hypotheses, and this should be done.

Reviewer 1 ·

Basic reporting

Their revision has improved the manuscript, though there are still issues with the current version. While I certainly think the study is relevant and publishable, there are still areas of the manuscript that could be improved/revised.

Some minor editing is included in comments section below.

As pointed out by previous reviewer, misuse of basal group. This is a common mistake but one that has been highlighted by a number of authors and steps to prevent this misuse should be taken. Basal implies it is primitive which is misleading and this comes across in some of the interpretation of the phylogeny.

Crisp & Cook (2005) "Do early branching lineages signify ancestral traits?" TREE 20(3): 122-128
Krell & Cranston (2004) Which side of the tree is more basal? Syst Ent. 29 (3)
279–281

Experimental design

On a positive note, the morphological analysis is improved with the addition of available data.

The phylogenetic analyses/data was not changed with the authors defending their position in the response to the reviews. Taking into consideration that they are adding samples to available genetic data/phylogenetic hypothesis, this should be clearer in the text. Most of the studies referenced were based on the same locus, cytb, with the difference being the addition of new taxa/geographic sampling so while it is it is good to see that the addition of taxa/samples doesn't flip the relationship around, these are not independent lines of evidence. They are just building on one another. The focus of Ingram et al and Davies et al were at the generic level which are supported both mt and nuclear data. While these different datasets are congruent for the relationships among genera, that does not provide evidence that mt gene tree = species tree.

Line 562 remove "extensive" . Phylogeographic analysis should include hypothesis testing. There are number of methods that could be included to test alternative hypotheses about phylogeographic history. No hypothesis testing was included.

Ree RH, Smith SA (2008) Maximum-likelihood inference of geographic range evolution by dispersal, local extinction, and cladogenesis. Syst Biol 57: 4–14.

Ronquist F (1997) Dispersal-vicariance analysis: a new approach to the quantification of historical biogeography. Syst Biol 46: 195–203.

Validity of the findings

The data supports the recognition of these two disjunct populations as distinct lineages worthy of species descriptions.


The paragraph on chromosomal evolution appears out of place since this isn't included in data/analysis. Should be removed.

Additional comments

Line 89 remove “fully”

Line 88-92 Awkward description

Line 122 remove “a standard range”

Line 266 remove “robust”

Line 501 – 506 Remove or rewrite. As currently written this is misleading.

Line 562 remove extensive, only a single locus. No hypothesis testing included.

Line 592 remove “strong”, while these populations provide additional data points you are building on the same data since this is all based on a single mt locus.

Need to check throughout body and figures for consistency of MTJ, ex. Line 305 & Fig 3 legend - MJT should be MTJ

Reviewer 2 ·

Basic reporting

Again, figures and supplementary data should be a bit modified, and descriptions and discussion slightly expanded considering morphological differences between adjacent populations. Some suggestions appear in the pdf file.

Experimental design

The author should explain clearly in the main text and for non-specialists the interest of using different phylogenetical methods. They should also emphasize the interest of studying skull shape, because these data are neither included in the description of species, nor discussed. It is also important to describe and discuss the differences observed between male and females, if they chose to display them in plots (figure 4).

Validity of the findings

Geometric morphometric results still need to be confirmed by statistical tests. Morphological differences between species (and sexes) need to be more accurately considered and discussed.

Additional comments

I recognize that the authors made significant efforts to improve their work, by adding new data. However, they should still improve some methodological and didactic aspects, as well as some points detailed in the pdf file, especially a few issues they did unfortunately not take into account after the first review.
I hope that these last suggestions will help the authors to improve their manuscript.

Annotated reviews are not available for download in order to protect the identity of reviewers who chose to remain anonymous.

---

## Round 0.3 · accepted · Accept

I appreciate the thoroughness to which you incorporated reviewer comments, and I believe the more rigorous morphological and phytogeographic analyses and discussions make for a stronger paper overall. Thank you for considering PeerJ for your manuscript, and also, please consider making your peer reviews open.